# A General Equilibrium Analysis of Achieving the Goal of Stable Growth by China's Market Expectations in the Context of the COVID-19 Pandemic

Jin Fan [1],*, Hongshu Wang [1] and Xiaolan Zhang [2]

1   College of Management and Economic, Nanjing Forestry University, Nanjing 210037, China
2   Division of Economic Forecasting, State Information Center of China, Beijing 100045, China
*   Correspondence: jfan@njfu.edu.cn

**Abstract:** The COVID-19 pandemic triggered a weakening of expectations of market players and local governments. It is necessary to explore some effective paths to stabilize China's market expectations. This paper draws on computable general equilibrium (CGE) model elasticity indicators and marginal utility indicators to simulate the impact of the COVID-19 pandemic on market agents and the impact of shocks brought about by the COVID-19 pandemic on various industry sectors. Our results first show that the Keynesian closure is still valid, with the impact values of the COVID-19 pandemic on GDP, urban consumption, rural consumption, investment, imports, and exports being 2.35%, 7.96%, 9.79%, 4.10%, −3.13%, and 6.15%, respectively, with the COVID-19 pandemic shock having a particularly strong impact on rural consumption. Second, the impact of the COVID-19 pandemic on different industries is comprehensive and non-equilibrium. In consumption, urban and rural residents have the largest changes in consumption demand in the tertiary and primary industries, and the overall change in consumption of rural residents is larger than that of urban residents. In investment, the tertiary industry investment changes most significantly, while the primary industry has a gentle change. The impact of weaker import and export expectations on industry imports and exports is concentrated in the secondary and primary industries. Third, the weakening expectation induces a decline in the multiplier effect, making it difficult for economic growth to return to pre-pandemic levels, which needs to be accompanied by fiscal policies such as reducing taxes, lowering fees, and raising fiscal spending to achieve growth targets. In addition, while fiscal policy significantly boosts import and export trade, it has a significantly greater impact on exports than imports.

**Keywords:** COVID-19; market expectations; fiscal policy; CGE model

## 1. Introduction

The Central Economic Work Conference in 2021 first suggested that China's economy was facing the triple pressure of "demand contraction, supply shock, and weakening expectations", and in early March 2022, Premier Li Keqiang's "Government Work Report" again judged that China's economy was facing this "triple pressure". In the fourth quarter of 2021 and the first quarter of 2022, the national GDP (gross domestic product) grew by 4% and 4.8% year-on-year, respectively. This was significantly slower than the previous quarter and lower than the 5.5% target set at the beginning of the year, further validating the increased downward pressure on economic growth. The COVID-19 pandemic has led to some increasing risks and challenges, and the complexity, severity, and uncertainty of China's economic development environment have been highlighted, with new challenges to stabilize growth, employment, and prices [1]. The overlap of the centennial change and the COVID-19 pandemic has formed a weakening of expectations stemming from the cross-conduction of long-term and short-term factors [2,3], as evidenced by the following.

First, consumption expectation has weakened. The rise of the service sector as a proportion of GDP is one of the most important indicators of social progress, and the

COVID-19 pandemic first hit the consumer service sector, which is related to human contact and communication [4], and the productive service sector has been further affected since, while the continued impact of the emergence of the COVID-19 pandemic can be spread through logistics channels. Meanwhile, the ineffective and inefficient investment by state-owned enterprises and local governments on a huge scale creates demand in the current period but suppresses it in the long run, while some individuals with private capital appears reluctant or afraid to invest in the phenomenon. Therefore, the weakening of employment income expectations will inevitably lead to weaker consumption.

Second, investment expectation has gradually weakened. With the COVID-19 pandemic, residents have become more cautious in their daily lives, including in their investment behavior [5,6]. As a vicious circle, logic has been formed between industrial capital and financial capital for a long time: excess speculative capital is no longer interested in the real economy and manufacturing industry with increasingly low gross margins, while excessive capital investment brings price increases in raw materials, rents, etc., and then further squeezes industrial capital profits. In the second half of 2021, it was clear that the weakening of real estate expectations had further reinforced the weakening of investment expectations.

Third, import and export expectations have become weaker [7–9]. In the past three years, China's economy has endured some heavy impacts from the U.S.–China trade war, science and technology war, and the pandemic. In addition, the policy chain, industrial chain, talent chain, innovation chain, capital chain and ecological chain have been honed, but the world risk index brought about by the Sino–US trade war is higher than other events. Some countries acted independently on the basis of national security zones, which had a great impact on global value chains and supply chains [10].

Fourth, local government expectations have weakened [11]. Guangdong, Jiangsu, and Shandong, the three strongest provinces in China's economy, have set their economic growth targets for 2022 at around 5.5%. This actually indicates that local governments are expecting a weakening of the uncertainty caused by the COVID-19 pandemic. For many years, land finance has been one of the most important aspects of local GDP championships, but the weakening of real estate and the cost of controlling the "zero" epidemic have undoubtedly increased local financial difficulties, which have posed a huge challenge to the governance capacity of local governments. Overall, the COVID-19 pandemic, the income gap after China's GDP per capita crossed the upper limit of the "middle-income trap", industrial upgrading, government governance and other social changes, economic transformation, government effectiveness, and other factors constitute the main causes of the weakening expectations of market participants.

Since 1990s, the CGE (computable general equilibrium) model has been widely used as an effective policy analysis tool to analyze the possible impacts of China's accession to the WTO, environment, taxation, finance, pension insurance, enterprise reform, trade liberalization, carbon emissions, water management and other policy measures, and external shocks on China's macro and regional economies. It has also been widely applied to analyze the possible impacts of China's macro and regional economies, especially the impact of the COVID-19 pandemic on China's economy in recent years [12–15].

The macroeconomic impact of the COVID-19 pandemic has been studied in their respective fields. First, many pieces literature have discussed the impact of the COVID-19 pandemic on labor. Brada et al. [16] studied the resilience indicators of 199 countries in Central and Eastern Europe and pointed out that the COVID-19 pandemic would cause the recovery of labor employment to be very slow. Goswami et al. [17] claimed that the COVID-19 pandemic had caused the labor in the secondary and tertiary industries to suffer a greater degree according to the summary data of India. Mulligan [18] assessed the opportunity cost of work stoppage since the COVID-19 pandemic in the American System of National Accounts and inferred the welfare loss caused by non-working days and the decline of labor–capital ratio caused by layoffs.

In addition, some researchers have investigated the impact of the COVID-19 pandemic on financial markets from the perspective of the financial markets. Haroon et al. [19] claimed that the outbreak of the COVID-19 pandemic had caused instability in the financial markets and led to price fluctuations. Carlsson et al. [20,21] simulated the indirect impact of the COVID-19 pandemic on the financial markets and the impact on the real economy through scenario analysis. Elenev et al. [22] claimed that the decline in productivity and labor supply caused by the COVID-19 pandemic indirectly led to the bankruptcy of financial intermediaries.

Finally, the impact of the COVID-19 pandemic on business demand has not been ignored. Balla et al. [23] claimed that the COVID-19 pandemic had caused the temporary closure of millions of American enterprises. Through demand forecasting, they found that small enterprises would delay the opening of companies after the blockade. Chetty et al. [24] studied the bankruptcy of enterprises and believed that it was reasonable to keep the business delay or closure caused by the blockade within 15%.

Macro closure (macroeconomic closures, closure), which originated from Sen (1963), is an indispensable selection step for CGE model to solve equilibrium solutions. The process actually implies the model users' preferences for different theories, corresponding to different theoretical schools of interpretation of macroeconomic phenomena and corresponding policy measures [25–30]. Taylor [31] argues that "Macroclosure selection is crucial, and arbitrary choices may have southward consequences." Its doctrinal logic addresses the phenomenon of over determination in macroeconomic models, i.e., when the number of endogenous variables and equations in an economic system model is not equal, some of the equations need to be deleted or some of the endogenous variables need to be transferred to exogenous variables to achieve a solution.

Different macroeconomic schools of thought have given very different interpretations of the mechanisms and governance solutions for dealing with the economic crisis. Representative views include: New Keynesianism advocates cutting taxes and fees, lowering the macro tax burden, increasing disposable income and corporate profits, driving consumption and corporate investment; the supply school of thought advocates defining the relationship between the government and the market, deregulating the market, reducing administrative monopoly, defining the responsibility for affairs and expenditure, streamlining the government, improving human capital, increasing the efficiency of market allocation of production factors, and thus increasing total factor productivity; the neoclassical school of thought advocates establishing a market-oriented competition mechanism, creating a low-cost business environment, and restoring corporate confidence; the monetarist school of thought advocates improving the foresight of monetary policy, maintaining a reasonable abundance of liquidity, preventing another release of stimulus, and relying mainly on savings to drive investment [32–36].

Faced with the uncertainties of multiple factors, such as the Sino–US trade war friction and the COVID-19 pandemic, the idea of using macro closure to screen different macroeconomic views is helpful to provide information on China's exploration and typical examples that are familiar with international mainstream economics for the implementation of factors "based on the new development stage, implementation of the new development concept, and building a new development pattern". In particular, it needs to be studied at the level of structure and mechanism, which is of great practical significance and theoretical value for enriching and developing the theoretical system of socialist political economy with Chinese characteristics.

The innovations of this paper: under the framework of general equilibrium analysis, the first innovation is to provide a macroeconomic theory more consistent with the explanation of reality by comparing exogenous shocks with real data with the help of macro-closure analysis; the second is to show the shift of market agents' expectations through the change of economic elasticity; the third is to present an effective path to stabilize market expectations with the help of scenario analysis. The paper is organized as follows: in Section 2 we

will provide some theoretical models. Some empirical tests will be presented in Section 3. Section 4 is a scenario simulation. Finally, some policy suggestions will be given.

## 2. Theoretical Model and Data

### 2.1. Standard CGE Model

In this paper, the IFPRI (The International Food Policy Research Institute) standard CGE model of Lofgren [37] was chosen mainly based on the following considerations: on one hand, the means of the model's macro closure are more flexible and convenient, and multiple combinations of closures (theoretically 120 different closures) can be realized. Traoré (2012) counts 90 closures in the IFPRI standard CGE model due to the lack of differentiation between the elements in the model section. In contrast, the GAMS procedure, part of the IFPRI standard CGE model of Lofgren et al. (2002), divides the elements into labor, land, and therefore 120 species.

The IFPRI standard CGE model consists of a price module, a production and trade module, a sector module, and a system constraints module (macro closure), with a total of 48 classification equations.

### 2.2. Elasticity Parameters Reflecting Market Expectations in the CGE Model

#### 2.2.1. Consumption Expectation and Consumption Elasticity

Consumption expectation refers to the marginal propensity to consume made by consumption subjects in response to market economic conditions, which is not only influenced by market conditions but also inextricably linked to income and prices. In general, income elasticity, price elasticity, and consumption elasticity can be used to simulate consumption expectation scenarios. The IFPRI standard CGE model uses the marginal propensity to consume in simulating the weakening of consumption expectation, which indicates the expansion and contraction of the consumer subject's demand for a given commodity. In the standard CGE model, there are 20 variables, 12 exogenous givens, 3 calibrated by the SAM table, and 5 jointly determined by the SAM (social accounting matrix) and exogenous parameters. Among the variables reflecting consumption expectations are $\beta_{ch}^{m}$ and $\gamma_{c\,h}^{m}$. $\beta_{ch}^{m}$ represents the marginal share of consumption expenditure of household h for market commodity c, and $\gamma_{c\,h}^{m}$ represents the per-capita subsist consumption of household h for market commodity c. Both of them vary through consumption elasticities, as shown in Equations (1) and (2).

$$\beta_{ch}^{m} = BUDSHR(C,H) \times LESELAS(C,H) \tag{1}$$

where $BUDSHR(C,H)$ refers to the budget share of household h for commodity c; $LESELAS(C,H)$ refers to the expenditure elasticity of household h for commodity c.

$$\gamma_{c\,h}^{m} = \begin{array}{l} ((SUM(CP,\ SAM(CP,H) + SUM(AP,\ SAM(AP,H))) \ /\ PQ0(C)) \\ \times \big(\ BUDSHR(C,H) + \beta_{ch}^{m}/FRISCH(H)\big) \end{array} \tag{2}$$

where $CP$ refers to the commodity set; $AP$ refers to the activity set; $SAM(CP,H)$ refers to the number of households h in the SAM table for the commodity set; $SAM(AP,H)$ refers to the number of households h in the SAM table for the activity set; $PQ0(C)$ refers to the price of the composite commodity c; and $FRISCH(H)$ refers to the Frisch parameter.

#### 2.2.2. Investment Expectation and Investment Elasticity

Investment expectation refers to the investment subject's view and judgment on the future market investment behavior. Using investment elasticity can reflect the impact on economic growth and investment behavior after the confidence of investment subjects becomes weak due to the COVID-19 pandemic. In the IFPRI standard CGE model, the variables reflecting investment elasticity are $\rho_a^{va}$ and $\alpha_a^{va}$. $\rho_a^{va}$ denotes the CES activity production function exponent, and $\alpha_a^{va}$ denotes the shift parameter for CES activity produc-

tion function. The specific formulas are shown in Equations (3) and (4). $\rho_a^{va}$ varies in the opposite direction from the investment elasticity, while $\alpha_a^{va}$ varies through the variable $\rho_a^{va}$.

$$\rho_a^{va} = (1/PRODELAS(A)) - 1 \tag{3}$$

where $PRODELAS(A)$ refers to the elasticity of substit factors.

$$\begin{aligned}\alpha_a^{va} = \ & QVA0(A)/(SUM(F\$(QF0(F,A)), deltava(F,A) \times QF0(F,A) \\ & \times (-rhova(A)))) \times (-1/rhova(A))\end{aligned} \tag{4}$$

where $QVA0(A)$ refers to the quantity of total value added of activity a; $QF0(F, A)$ refers to the demand of activity a for factor f; $deltava(F, A)$ refers to the shared efficiency parameter of the CES value added function of factor f in activity a.

### 2.2.3. Import Expectation and Armington Elasticity

Import expectation reflects the forecasts of market subjects in foreign trade regarding the import situation. In this paper, the Armington elasticity of substitution in the IFPRI standard CGE model is selected to reflect the strength of the impact of relative international prices on changes in demand for imported products, which is an important parameter for understanding the characteristics of the global economy. The larger the Armington elasticity of substitution, the stronger the relationship of substitution between national products and imports, and then the higher the cost of imports will rise due to port blockages and labor shortages caused by the COVID-19 pandemic. Thus, the demand for imported products will fall sharply. In the standard model, the variable $\rho_c^q$ is used to reflect the Armington function exponent, as shown in the following Equation (5).

$$\rho_c^q = (1/TRADELAS(C, SIGMAQ)) - 1 \tag{5}$$

where $TRADELAS(C, SIGMAQ)$ denotes the Armington elasticity of commodity c.

### 2.2.4. Export Expectation and CET Elasticity

Export expectation corresponds to import expectation and indicates the judgment of economic agents on the export situation. Under the pressure of weakening export expectation, the CET (constant elasticity of transformation) elasticity can be used to study the impact of falling export expectations on economic growth. The CET elasticity of substitution is used to reflect the relationship between domestic consumption and exports. The larger the CET elasticity, the stronger the substitution relationship between domestic consumption and exports. When prices fall, the exports will gradually diminish in quantity. Here the exogenous variable $\rho_c^t$ is selected in the IFPRI standard CGE model as the intrinsic correlation of the change in CET elasticity, and the specific formula is shown in Equation (6).

$$\rho_c^t = (1/TRADELAS(C, SIGMAT)) + 1 \tag{6}$$

where $TRADELAS(C, SIGMAT)$ denotes the CET elasticity of commodity c.

### 2.3. Data

The IFPRI standard CGE model data for China are derived from SAM based on national income accounts, input–output tables, cash flow tables, and other relevant statistics. SAM is a comprehensive overall economic data framework that captures a country's economy. It is a square matrix in which each account can be represented by its rows and columns. Each element of the matrix represents the payments from the column accounts to the row accounts. Table 1 is based on Lofgren et al. [37] and Fanjin et al. [25], whereby the preparation process and data source can be found in Fanjin et al. [25].

**Table 1.** China Macro SAM (2018) (unit: in hundreds of billions, Yuan).

| | Comm-odities | Acti-vities | Labor | Capital | House-holds | Enter-prises | Government Subsidies | Extra-Budgetary Institutional | Gover-nment | The Rest of the World | Savings/ In-vestments | Stock Change | Total |
|---|---|---|---|---|---|---|---|---|---|---|---|---|---|
| Commodities | | 1573.44 | | | 347.36 | | | 20.19 | 128.21 | 164.13 | 407.71 | 39.96 | 2681.01 |
| Activities | 2495.50 | | | | | | | | | | | | 2495.50 |
| Labor | | 475.03 | | | | | | | | | | | 475.03 |
| Capital | | 350.71 | | | | | | | | | | | 350.71 |
| Households | | | 475.03 | 53.24 | | 15.30 | 0.31 | | 4.98 | 1.55 | | | 550.41 |
| Enterprises | | | | 296.05 | | | | | | | | | 296.05 |
| Government-Subsidies | | −16.33 | | | | | | | 16.63 | | | | 0.31 |
| Extra-budgetary institutional | | 70.19 | | | | | | | | | | | 70.19 |
| Government | 19.73 | 42.45 | | | 13.87 | 39.17 | | | | 19.26 | 149.61 | | 284.09 |
| The rest of the world | 165.79 | | | 1.42 | | | | | 4.60 | | | | 171.81 |
| Savings/Investments | | | | | 189.18 | 241.58 | | 50.00 | 129.65 | −13.13 | | | 597.28 |
| Stock change | | | | | | | | | | | | 39.96 | 39.96 |
| Total | 2681.01 | 2495.50 | 475.03 | 350.71 | 550.41 | 296.05 | 0.31 | 70.19 | 284.09 | 171.81 | 597.28 | 39.96 | |

Note: data were obtained from the authors' calculations.

The reason why this study chose the 2018 China SAM as the benchmark data is based on the fact that, on the one hand, the US–China trade friction started in March 2018, when then US President Donald Trump officially signed a trade memorandum with China in the White House, proposing that tariffs would likely be imposed on $60 billion of commodities imported from China and restricting Chinese companies from investing in M&A in the US; on the other hand, the COVID-19 pandemic began in early December 2019 with a dozen cases of a new type of pneumonia in Wuhan, Hubei, which became clear around 11 December. The 2018 SAM is the unaffected dataset. The eight subject accounts in this dataset are activities, commodities, factors, households, enterprises, government, investments and savings, and foreign. The data are obtained from the 2018 Chinese non-competitive input–output tables, the Chinese financial flows tables, the Chinese fiscal yearbook, the Chinese customs yearbook, and other relevant statistical information. This macro SAM has the following features: first, it distinguishes between activities and commodities; second, it considers transaction costs; third, it distinguishes the government into core government as well as various tax indicators; fourth, it separates non-governmental organizations as an account independent from households and firms; and fifth, it divides household consumption into market transactions and self-consumption.

The Chinese micro-SAM, which is needed for the sectoral studies involved in the latter study, is mainly decomposed using a top-down decomposition, and split according to the 2018 Chinese non-competitive input-output ratio, and the balance is achieved using an integrated RAS and cross-entropy approach.

## 3. Calibration of Macroscopic Closure and Benchmark Parameters

### 3.1. Macro Closure

#### 3.1.1. Selection of Key Macro Data

In this paper, the 2018 GDP, consumption, investment, and net export data published by the National Bureau of Statistics are used as the basic macro data, while the annual and quarterly data from 2012–2021 are referred to as the control data. Figure 1 shows the year-on-year growth rates of GDP, consumption, and investment from 2012 to 2021. We need to point out that, compared with the other variables, the growth rate of net export changes vary significantly, thus it is not shown in Figure 1 and it is illustrated by text instead. After 2018, the year-on-year GDP growth rate started to decline from a stable 6–7%, especially in 2020, when the year-on-year GDP growth rate fell to 2.35% due to the COVID-19 pandemic, but it rebounded to 8.10% in 2021. In terms of consumption and investment growth rates, the year-on-year growth of investment is relatively flat, and the consumption growth rate fluctuates greatly. In 2021, the growth rate of net export increased from −53.24% in 2018 to 773.67%, and its variation is much higher than consumption and investment, which shows that the impact of the COVID-19 pandemic on the net export growth rate is much higher than consumption and investment.

In the ring growth, this paper is based on the quarterly data of March 2018 to March 2022. Figure 2 shows the ring growth rates of various indicators, which are the same as Figure 1, and the net export data excluded are explained by text instead. The ring growth rate of GDP in the first quarter of 2020 is −26.03%, and the ring growth rates of consumption and investment are −24.48% and −58.71%, respectively, which are much lower than the growth rate of the same period. This period is the early stage of the COVID-19 pandemic outbreak, and the domestic economy was in a significant downward cycle, economic growth pressure was high, and the troika driving the economy was in a downturn. At the same time, the trade deficit hit a record high for the same period. In March 2022, China experienced a second round of the pandemic, and GDP, consumption, and investment faced downward pressure again, but the downward magnitude was lower than in the first quarter of 2020. The growth rate of net export fell to −6572.64% in the first round of the pandemic, which was 268 and 113 times higher than the consumption and investment growth rate, respectively. In addition, in the second round of the pandemic, the net export growth rate fell to −88.24%, which was higher than the investment growth rate.

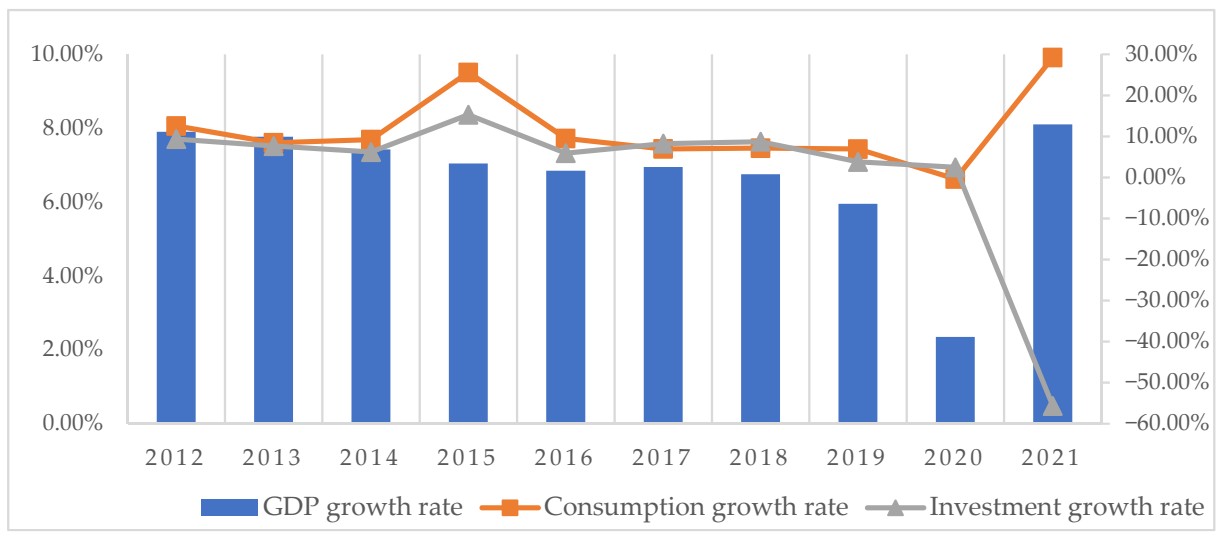

**Figure 1.** 2012–2021 year-on-year growth rates of GDP, consumption, and investment.

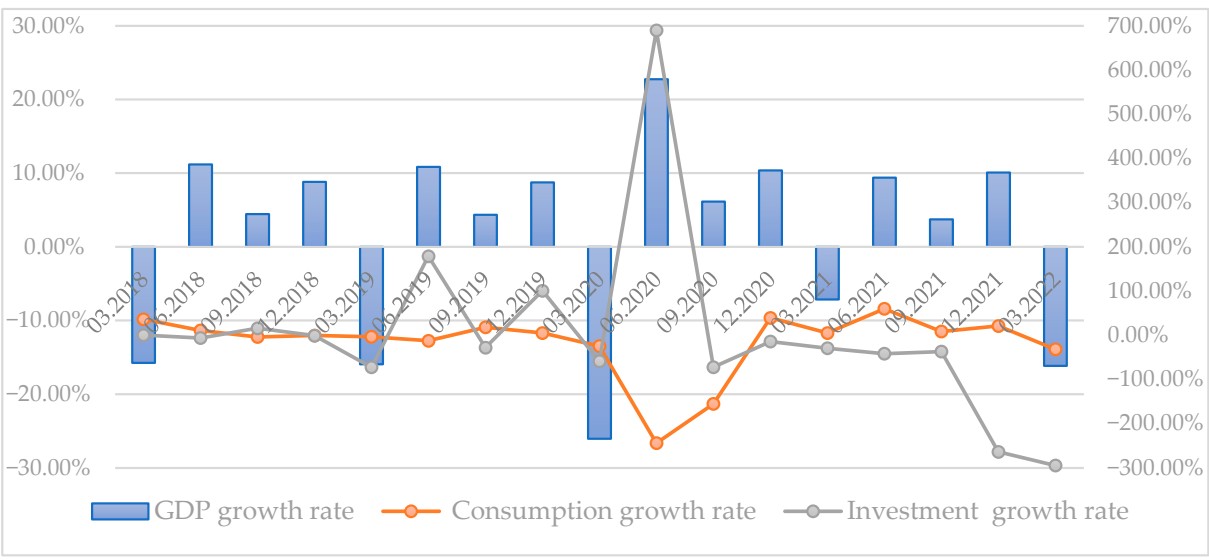

**Figure 2.** 2018–2022 ring growth rates of GDP, consumption, and investment.

The macro social accounting matrix is based on macro data, while in the micro social accounting matrix, the commodities and activities accounts are specifically classified. The 2018 input–output table consists of 142 economic sectors, and to simplify the analysis we divided the 142 sectors into 42 parts according to the national economic industry classification criteria to facilitate the subsequent study. At the same time, this paper analyzes the propensity of different consumption agents to cope with the expected weakening of consumption by dividing the residential accounts into two categories: urban and rural.

### 3.1.2. The Choice of Macroscopic Closure

In terms of macroscopic closure, this paper adopts Keynesian closure mainly based on two reasons: first, considering the characteristics of China's factor market, the labor market possesses a large amount of surplus labor due to the COVID-19 pandemic and there is a large amount of involuntary unemployment, so we cannot copy the neoclassical theory and assume full employment of labor to set macro closure. Second, most institutions and scholars believe that the impact of the COVID-19 pandemic on our economy should be short-term, which is in line with the short-term macroeconomic structure reflected by the Keynesian closure. The core of Keynesianism revolves around two assumptions: one is that

the currency is non-neutral, and the other is that the market is imperfect. Keynesianism is more from the demand side, while the supply chain is from the supply side. Keynesian closure has the following characteristics: (1) the level of investment and savings rates are fixed, and there is a lag in the adjustment of savings rates, which can lead to difficulties in achieving a balance between social investment and savings, (2) factor prices have the trouble of falling, and households need to put in more labor to ensure income growth, which in turn leads to an increase in employment and a decrease in wages, and (3) real output is determined by effective demand.

*3.2. Calibration of Benchmark Parameters*

The benchmark parameters are mainly set with reference to the values taken from the GTAP 10th edition China database [38], where the consumption elasticity of residents for different commodities is between 0.9–1.5, the investment elasticity is between 0.5–1.25, the Armington elasticity is between 0.5–3, and the CET elasticity is between 0.375–1.5, and the various elasticity parameters are based on 42 sectors for derivation and division, and the 42 sectoral elasticities are shown in Table A1, Appendix A.

3.2.1. Consumption Expectation and Consumption Elasticity

By changing the baseline parameters of consumption expectation, consumption sensitivity analysis can be conducted to determine the baseline parameters of this paper in the context of the COVID-19 pandemic, as shown in the later paper for investment, import, and export simulation ideas, which are uniformly described here.

Eight combinations of consumption expectations are assumed to change. Table 2 shows the results of the consumption sensitivity analysis, including the percentage change of the equilibrium solution of GDP, urban and rural consumption from the benchmark value after simulation. Various changes are described in the table, where URBANHH is the urban marginal propensity to consume and RURALHH is the rural marginal propensity to consume.

**Table 2.** Consumption sensitivity analysis (unit: %).

| | GDP | Urban Consumption | Rural Consumption |
|---|---|---|---|
| **Change 1:** URBANHH reduced by 1.25 times, RURALHH reduced by 1.25 times | −4.75% | −8.38% | −10.21% |
| **Change 2:** URBANHH enlarged by 1.25 times, RURALHH reduced by 1.25 times | −2.34% | −7.96% | −9.79% |
| **Change 3:** URBANHH enlarged by 1.2 times, RURALHH reduced by 1.2 times | −1.84% | −4.55% | −5.42% |
| **Change 4:** URBANHH reduced by 1.2 times, RURALHH enlarged by 1.2 times | 0.44% | 0.37% | 1.40% |
| **Change 5:** URBANHH reduced by 1.15 times, RURALHH enlarged by 1.15 times | 0.34% | 0.29% | 1.31% |
| **Change 6:** URBANHH enlarged by 1.15 times, RURALHH reduced by 1.15 times | −1.37% | −4.15% | −5.05% |
| **Change 7:** URBANHH reduced by 1.1 times, RURALHH enlarged by 1.1 times | −0.33% | −0.27% | −1.13% |
| **Change 8:** URBANHH enlarged by 1.1 times, RURALHH reduced by 1.1 times | −0.95% | −1.80% | −3.73% |

Note: data were obtained from the authors' calculations.

As can be seen in Table 2, Change 7 is closest to the base case, with a change in GDP of −0.33%. Change 2 can be used to simulate the impact on consumption due to the COVID-19 pandemic, when the change in GDP is similar to the GDP growth rate of 2.35% announced by the National Bureau of Statistics in 2020, with urban and rural consumption falling by 7.96% and 9.79%, respectively. When the consumption subject is pessimistic about consumption behavior, that is, 1.2 times of the marginal propensity to consume in rural areas in Change 3, this will lead to a 1.84% drop in GDP; when reversing expectations

and increasing consumption elasticity, such as in Change 5, rural marginal propensity to consume enlarged by 1.15 times and GDP grew by 0.34%; when consumption expectations are very optimistic, rural marginal propensity to consume continued to expand to 1.2 times and GDP grew by 0.44%. A rise in urban residents' marginal propensity to consume leads to weaker consumption expectations, while rural residents' marginal propensity to consume acts as a positive pull on consumption expectations. Moreover, it can be seen that the key to reverse residents' expectations lies in stimulating rural residents' domestic demand and enhancing rural residents' main confidence.

### 3.2.2. Investment Expectation and Investment Elasticity

Eight changes in investment are assumed, the specific results are shown in Table 3.

**Table 3.** Investment sensitivity analysis (unit: %).

| | GDP | Fixed Investment | Stock Change | Account: Total | Household | Government |
|---|---|---|---|---|---|---|
| **Change 1:** investment elasticity enlarged by 1.25 times | 0.01% | 0.19% | 0.02% | 0.02% | 0.01% | −0.04% |
| **Change 2:** investment elasticity enlarged by 1.5 times | 0.51% | 0.29% | 0.03% | 0.03% | 0.02% | −0.07% |
| **Change 3:** investment elasticity enlarged by 2.5 times | 2.33% | 3.57% | 1.06% | 1.05% | 1.04% | −1.134% |
| **Change 4:** investment elasticity enlarged by 5 times | 2.45% | 3.75% | 2.07% | 2.06% | 2.05% | −2.18% |
| **Change 5:** investment elasticity enlarged by 8 times | 2.550% | 3.81% | 2.08% | 2.07% | 2.06% | −2.20% |
| **Change 6:** investment elasticity reduced by 1.25 times | −0.02% | −1.20% | −0.02% | −0.03% | −0.02% | 0.06% |
| **Change 7:** investment elasticity reduced by 2.5 times | −1.12% | −3.68% | −1.16% | −1.03% | −1.02% | 1.08% |
| **Change 8:** investment elasticity reduced by 3 times | −2.35% | −4.10% | −2.20% | −2.20% | −2.15% | 2.52% |

Note: data were obtained from the authors' calculations.

Among investment expectation, Change 1 is closest to reality, when the investment elasticity enlarged by 1.25 times, and GDP grew by 0.01%. Change 8 is in line with the expected change of the COVID-19 pandemic, and fixed investment dropped by 4.10%. When the investment subject is extremely pessimistic, the elasticity reduced by 2.5 times, GDP dropped by 1.12%; when the investment subject is pessimistic, GDP dropped by 0.02%; when the investment subject is more optimistic about investment behavior, the elasticity enlarged by 2.5 times, GDP grew by 2.33%; when the investment body is very optimistic, the elasticity enlarged by 8 times, GDP grew by 2.55%. This reflects that investment expectations grow in the same direction when the elasticity of investment becomes larger. In addition, the weakening of consumption expectation is manifested in a lower marginal propensity to consume, which is directly related to the investment multiplier. The higher the marginal propensity to consume, the higher the investment multiplier, which coincides with the expected turn in the same direction as that reflected by the marginal propensity to consume and the elasticity of investment of rural residents.

### 3.2.3. Import Expectation and Armington Elasticity

Regarding the sensitivity analysis of import and export expectations, eight import- and export-simulated portfolio changes are selected in this paper. The specific results are shown in Table 4.

**Table 4.** Import and export sensitivity analysis (unit: %).

|  | GDP | Import | Export | Foreign Account |
|---|---|---|---|---|
| **Change 1:** Armington elasticity enlarged by 1.25 times and CET elasticity enlarged by 1.5 times | 0.06% | 0.42% | 1.02% | 0.42% |
| **Change 2:** Armington elasticity enlarged by 1.5 times and CET elasticity enlarged by 1.25 times | 0.26% | 1.06% | 1.26% | 0.07% |
| **Change 3:** Armington elasticity reduced by 1.25 times and CET elasticity enlarged by 1.25 times | 0.02% | 1.10% | 2.10% | 0.19% |
| **Change 4:** Armington elasticity enlarged by 1.25 times and CET elasticity reduced by 1.25 times | 0.20% | 1.17% | 1.18% | 0.17% |
| **Change 5:** Armington elasticity reduced by 2 times and CET elasticity reduced by 2 times | −2.34% | −3.13% | 6.15% | 2.13% |
| **Change 6:** Armington elasticity reduced by 5 times and CET elasticity reduced by 5 times | −2.38% | −3.49% | −7.50% | −2.87% |
| **Change 7**: Armington elasticity reduced by 8 times and CET elasticity reduced by 8 times | −3.88% | −3.83% | −7.84% | −3.83% |
| **Change 8:** Armington elasticity reduced by 10 times and CET elasticity reduced by 10 times | −4.01% | −3.06% | −8.07% | −4.05% |

Note: data were obtained from the authors' calculations.

Among import expectations, for Change 3, the Armington elasticity reduced by 1.25 times, which is closest to the baseline case, with an import-to-baseline differential ratio of 1.10% and a GDP differential ratio of 0.02%. Comparing Change 1 and 2 together, we can see that GDP grew by 0.06% when the Armington elasticity enlarged by 1.25 times and by 0.26% when the Armington elasticity enlarged by 1.5 times. It can be seen that the greater the Armington elasticity of substitution, the greater the import expectations reflected.

### 3.2.4. Export Expectation and CET Elasticity

According to the results of the sensitivity analysis of imports and exports obtained in Table 4 above, the CET elasticity is most consistent with the baseline scenario when it is enlarged by 1.25 times and the export differential is 2.10%. Change 5: both Armington and CET elasticities reduced by 2 times and they can be used to simulate economic growth resulting from the COVID-19 pandemic. As a result of the COVID-19 pandemic, logistics issues have caused blockages, resulting in the inability to trade exports. Assuming that the export expectation is pessimistic, when the CET elasticity reduced by 2, 5, and 10 times, GDP dropped by 2.34%, 2.38%, and 4.01%, respectively. When the expectation was reversed, the CET elasticity enlarged by 1.25 times and GDP grew by 0.02% and when it would continue to grow with confidence, the elasticity enlarged by 1.5 times and GDP grew by 0.06%. As can be seen, the same as for the Armington elasticity, the greater the CET elasticity, indicating the greater the export expectations, the more significant the pull on economic growth. In addition, the overall impact of weakening expectations due to the COVID-19 pandemic on exports is greater than on imports. The reason for this could be that exports are more deeply affected by direct supply-side shocks, while imports are mainly affected by the overall contraction of the economic scale.

### 3.3. Test Results

Since consumption, investment, import- and export-related variables are included in the sensitivity analysis, the GDP indicator is chosen to test the results. In this paper, the ARIMA (autoregressive integrated moving average) model in time series analysis is used to analyze the values of China's GDP from 2002–2017, which in turn predicts the GDP growth trend in 2020, and the result obtained is −4.5%. Combined with the sensitivity analysis above, it was observed that when the marginal propensity to consume enlarged by 1.25 times in urban consumption, GDP reduced by 1.25 times in rural consumption, reduced by 3 times in investment elasticity, reduced by 2 times in Armington elasticity, and reduced by 2 times in CET elasticity. This is consistent with the expected weakening scenario due

to the COVID-19 pandemic, thus determining the elasticity parameters selected in this paper on the elasticity parameter benchmark, and the specific results are shown in Table A2, Appendix B.

After confirming the baseline elasticity parameters, the 42 sectors are compared with the base period values on the basis of the new equilibrium solutions for consumption, investment, imports, and exports. It is not difficult to analyze the deviations of the impact of different economic behaviors on each sector under the expected weakening, whose results are shown in Table 5. The impact of the COVID-19 pandemic on industry is all-encompassing, even if some sectors are not directly affected in investment, but also through inter-industry transmission to consumption, imports, and exports.

**Table 5.** Simulation of sectoral consumption, investment, import, and export shocks (unit: %).

| Sector | Urban Consumption | Rural Consumption | Investment | Import | Export |
|---|---|---|---|---|---|
| Agriculture, forestry, animal husbandry, fisheries and service products | 17.59% | 19.12% | −4.18% | 9.11% | 21.48% |
| Coal mining products | 7.35% | 9.42% | — | 1.03% | 17.36% |
| Oil and gas extraction products | 8.34% | 9.71% | — | 5.46% | −11.66% |
| Metal mining products | 8.34% | 9.71% | — | 1.00% | −1.22% |
| Non-metallic and other ore mining products | 6.34% | 12.68% | — | 1.14% | 0.00% |
| Food and tobacco | 7.50% | 25.31% | — | −1.15% | 5.19% |
| Textiles | 7.40% | 9.81% | — | −6.73% | −1.13% |
| Textiles, clothing, shoes, hats, leather, down and other products | 8.40% | 7.87% | — | −1.34% | 8.14% |
| Woodworking and furniture | 8.39% | 6.87% | 3.18% | 7.13% | −1.03% |
| Paper printing and stationery and sporting commodities | 8.39% | 9.93% | 1.18% | −1.13% | 2.10% |
| Petroleum, cooking products and processed nuclear fuel products | 11.40% | 23.99% | — | −2.37% | 15.08% |
| Chemicals | 7.35% | 9.98% | — | −1.13% | 7.03% |
| Non-metallic mineral products | 8.41% | 8.02% | — | −5.27% | 12.62% |
| Metal smelting and rolling products | 10.36% | 8.02% | — | −2.12% | 7.13% |
| Metal products | 8.38% | 13.92% | 3.18% | −1.05% | 13.89% |
| General purpose equipment | 8.39% | 9.02% | 1.20% | −1.12% | 8.44% |
| Specialized equipment | 7.40% | 10.97% | 4.16% | 5.36% | 5.22% |
| Transportation equipment | 8.39% | 10.00% | 2.15% | −1.02% | 4.02% |
| Electrical machinery and equipment | 7.37% | 9.00% | 1.18% | −9.08% | 5.40% |
| Communication equipment, computers and other electronic equipment | 9.37% | 10.93% | 1.18% | 1.23% | 1.24% |
| Instrumentation | 7.40% | 10.88% | 1.18% | −1.07% | 2.29% |
| Other manufacturing products and scrap waste | 8.38% | 9.12% | — | −2.17% | 3.14% |
| Metal products, machinery, and equipment repair services | 11.46% | 10.09% | 2.18% | 1.23% | 3.89% |
| Electricity, heat production and supply | 8.39% | 12.08% | — | 1.00% | 0.00% |
| Gas production and supply | 9.41% | 10.04% | — | 1.06% | 8.06% |
| Water production and supply | 8.42% | 12.04% | — | 3.02% | 8.09% |
| Architecture | 9.36% | 12.83% | 4.17% | 3.57% | 9.06% |
| Wholesale and retail | 11.38% | 18.75% | 21.18% | −1.01% | 2.09% |
| Transportation, storage and postal | 25.91% | 19.82% | 29.18% | −1.13% | 5.62% |
| Accommodation and dining | 24.52% | 18.85% | — | −1.12% | 0.00% |
| Information transmission, software and information technology services | 7.34% | 11.89% | 8.16% | −4.30% | −3.16% |
| Finance | 7.37% | 13.89% | — | −2.64% | −1.20% |
| Real estate | 8.34% | 3.00% | 17.18% | −1.38% | 3.23% |
| Leasing and business services | 8.35% | 8.89% | — | −3.10% | 2.00% |
| Research and experimental development | 6.35% | 8.87% | 8.20% | −1.22% | 4.11% |
| Integrated technical services | 5.35% | 8.87% | 8.18% | −5.21% | −8.09% |

**Table 5.** *Cont.*

| Sector | Urban Consumption | Rural Consumption | Investment | Import | Export |
|---|---|---|---|---|---|
| Water, environment and utilities Management | 5.35% | 8.89% | — | −1.19% | 2.20% |
| Residential services, repairs and other services | 1.35% | 6.89% | — | 1.41% | 6.15% |
| Education | 4.35% | 6.89% | — | −2.32% | 2.16% |
| Health and social work | 2.33% | 4.89% | — | −1.46% | 2.23% |
| Culture, sports and recreation | 3.31% | 8.86% | 8.18% | −3.14% | 2.31% |
| Public administration, social security, and social organizations | 2.47% | 11.88% | — | −2.14% | 1.35% |

Note: data were obtained from the authors' calculations.

In general, the impact of the COVID-19 pandemic on the tertiary industry is the largest, and it is mainly reflected in the accommodation and dining, transportation, storage and postal, wholesale and retail, and real estate sectors. In order to minimize the flow of people, all localities strictly control travel, catering, and transportation. These industries are inevitably restrained, and the development of the industry is basically stagnant.

Specifically, in terms of consumption demand, urban residents first had the largest changes in consumption demand for transportation, storage and postal, accommodation and dining, and agriculture, while rural residents had the largest changes in consumption for food and tobacco, petroleum refinement, and agriculture. From the perspective of the three industries, the change in consumption for the rural residents for primary and secondary industries is greater than that for tertiary industries after the weakening of consumption expectations. In addition, in terms of the absolute value of changes, the overall change in consumption in rural areas is greater than that in urban areas. Thus, the key to reversing expectations is to expand the marginal propensity to consume of the rural middle- and lower-income classes.

Second, in terms of investment demand, the unevenness of the impact of the COVID-19 pandemic is reflected in the fact that the impact of the COVID-19 pandemic on investment in secondary industries is lower than that in primary and tertiary industries. This could be related to the "Carbon Neutrality and Emission Peak" development target, which constrains the production of high carbon-emitting industries by enterprises and thus reduces investment expectations. Therefore, it is necessary to strengthen investor confidence in such industries and promote economic growth within the scope of carbon emissions.

Third, in terms of import and export demand, the expected weakening of the industry's imports will have a deeper impact on agriculture, forestry, animal husbandry, fisheries and service products (9.11%), electrical machinery and equipment (−9.08%), woodworking and furniture (7.13%), textiles (−6.73%), and oil and gas extraction products (5.46%) sectors, while the expected weakening of the industry's exports will have a deeper impact on agriculture, forestry, animal husbandry, fisheries and service products (21.48%), coal mining products (17.36%), petroleum, cooking products and processed nuclear fuel products (15.08%), metal products (13.89%), and oil and gas extraction products (−11.66%). The export of agriculture, forestry, animal husbandry, fisheries and service products, oil and gas extraction, electrical machinery and other sectors changed significantly. In addition, the import of food, energy and timber was deeply affected by the COVID−19 pandemic. External uncertainty affects the import expectations of China, while having an impeding effect on economic growth.

After simulating the impact of the COVID-19 pandemic on consumption, investment, imports and exports, this paper uses the latest input–output table data from 2020 and 2018 to test the actual growth rates of consumption, investment, imports and exports against the simulated growth rates for errors. As can be seen from Table 6, the difference between the simulated and actual results is kept within a range of 1–2%. The trends of the aggregate changes in consumption, investment, imports and exports and the changes in the three

industries are consistent. We combine the actual results with the comparison of the errors of the simulated results to show that the simulated results are within a reasonable range.

**Table 6.** Simulated changes in consumption, investment, imports and exports (unit: %).

| | | Urban Consumption | Rural Consumption | Investment | Import | Export |
|---|---|---|---|---|---|---|
| Real growth rate | Total | 9.57% | 14.34% | 4.09% | −3.54% | 6.96% |
| | Primary Industry | 17.94% | 19.80% | −5.02% | 9.03% | 20.32% |
| | Secondary Industry | 10.20% | 12.69% | 2.00% | −0.29% | 5.52% |
| | Tertiary Industry | 8.24% | 14.13% | 16.78% | −2.12% | 1.43% |
| Simulated growth rate | Total | 7.96% | 9.79% | 4.10% | −3.13% | 6.15% |
| | Primary Industry | 17.59% | 19.12% | −4.18% | 9.11% | 21.48% |
| | Secondary Industry | 8.47% | 11.28% | 2.27% | −0.18% | 5.09% |
| | Tertiary Industry | 8.27% | 11.27% | 14.32 | −2.00% | 1.40% |

Note: data were obtained from the authors' calculations.

## 4. Scenario Analysis

### 4.1. Scenario Setting

According to the data disclosed by the National Bureau of Statistics, China's GDP growth rate at constant prices in 2020 was only 2.3%, much lower than 6% in 2019, and the GDP growth rate has slowed down significantly. As the domestic epidemic situation gradually improves and the impact of the COVID-19 pandemic on economic growth gradually weakens, the question of how to restore consumption, investment, import and export expectations to their original levels and to achieve an orderly improvement is an important issue in the current economic development. In view of this, this paper sets out three economic growth target scenarios and studies the effective paths to raise various expectations under different targets. Target I: based on the original level, the GDP growth rate returns to the original growth rate after excluding the disruption of the COVID-19 pandemic; Target II: based on the 5.5% target, the GDP growth rate returns to the target set out in the 2022 government work report; Target III: this target is based on the 5% target, which was set according to the 14th Five-Year Plan. The 14th Five-Year Plan calls for economic growth to remain within a reasonable range without setting a specific target value for economic growth, so this paper is based on the expected target of 5% predicted by Li Xuesong, director of the Institute of Quantitative and Technical Economics, Chinese Academy of Social Sciences.

### 4.2. Scenario Simulation

Fiscal policy, as an important macroeconomic tool for stimulating total domestic demand and raising expectations of consumption, investment, imports and exports, has an important pulling effect on stabilizing social development and promoting economic growth [39,40]. The common means to achieve fiscal policy are to reduce taxes and fees and raise government purchasing expenditures. In this paper, the effect of tax reduction is simulated by varying government transfer payments through the variable $transfr(INSD, GOV)$ and various taxes in the government accounts of the social accounting matrix.

#### 4.2.1. Raising Consumer Expectation

In terms of consumption expectations, since the outbreak of the COVID-19 pandemic, residents have been unable to travel and shop due to the impact of prices and income levels, and non-essential commodities such as transportation and restaurants have been significantly impacted, and residents' consumption expectation have been substantially reduced. Improving consumption expectations only by reversing these factors cannot bring GDP growth and urban and rural consumption back to their original levels and a policy needs to be implemented with active macro policies.

Table 7 shows the simulation results of consumption expectation under different fiscal policy combinations. It is important to note that different policy combinations are used under different objectives. Three policy combinations are set for Target I to choose the policy combination consistent with Target I, and three scenarios are set for Target II and III, i.e., no change in confidence, a decline in confidence, and a return to the original level of confidence, respectively, and fiscal policy combinations I, II, and III correspond to the impact on economic indicators under different scenarios.

**Table 7.** Simulation of different policy scenarios for raising consumption expectations due to the COVID-19 pandemic (unit: 10 million; %).

| | | Target I | | | Target II | | | Target III | | |
|---|---|---|---|---|---|---|---|---|---|---|
| | Initial Value | Policy I | Policy II | Policy III | Policy I | Policy II | Policy III | Policy I | Policy II | Policy III |
| GDP | 8,951,660 | 4.04% | 1.91% | 0.05% | 5.45% | 5.47% | 5.54% | 5.04% | 5.02% | 5.01% |
| Urban consumption | 2,582,200 | 3.79% | 3.77% | 0.97% | 5.55% | 5.27% | 4.04% | 5.06% | 5.02% | 5.39% |
| Rural consumption | 695,380 | 4.12% | 5.16% | 1.79% | 6.19% | 5.18% | 4.13% | 5.17% | 5.14% | 6.45% |
| Private consumption | 3,277,613 | 5.97% | 1.94% | 2.96% | 5.57% | 4.25% | 4.05% | 5.08% | 4.99% | 5.00% |

Note: data were obtained from the authors' calculations.

The three fiscal policy combinations for Target I are Policy I: 5% tax cut and 10% spending increase; Policy II: 5% tax cut and 5% spending increase; Policy III: 2% tax cut and 3% spending increase.

The three fiscal policy combinations for Target II are Policy I: 3% tax cut and 5.5% spending increase; Policy II: 5% tax cut and 6% spending increase; and Policy III: 3% tax cut and 4% spending increase.

The three fiscal policy combinations for Target III are Policy I: 6.5% tax cut and 5% spending increase; Policy II: 6.5% tax cut and 6% spending increase; and Policy III: 4% tax cut and 5% spending increase.

To achieve Target I, the effect of a 2% tax cut and 3% spending increase in Policy III can restore GDP growth to its original level, which is 0.05%. In Target II, the three expected scenarios result in GDP growth of 5.5% with urban consumption growing by 5.55%, 5.27%, and 4.04%, and rural consumption growing by 6.19%, 5.18%, and 4.13%, respectively. In Target III, the three expected scenarios make the GDP growth rate reach the 5% target when the urban consumption grew by 5.06%, 5.02%, and 5.39%, and the rural consumption grew by 5.17%, 5.14%, and 6.45%, respectively. It can be seen that the overall change of rural consumption under different policy combinations is higher than urban consumption, and the growth of residential consumption is higher than the growth of GDP. Rural consumption demand has a huge potential, and the key into tap it is to expand the income of the middle class, improve the consumption capacity, and open up the rural consumption market to promote economic growth. In addition, the fiscal policy required to achieve different targets differs. The lower the consumption expectation and the higher the GDP growth target, the more active the fiscal policy needs to be. For example, when consumption expectation falls, a 5% tax cut and a 6% spending increase are needed to reach the 5.5% growth target, and when confidence returns to its original level, a 3% tax cut and a 4% spending increase are needed. The lower the base, the more difficult it is to maintain the restored growth.

### 4.2.2. Raising Investment Expectations

The fiscal policy implemented for different targets under investment expectations is the same as consumption, and the effects on GDP, fixed investment, and stock change are shown in Table 8. Overall, the active fiscal policy plays a good role in the negative impact caused by the weakening expectation of China's investment body due to the COVID-19 pandemic. When investment expectation turns weaker, we should cooperate with the fiscal policy of a 2% tax cut and a 3% increase in spending to restore the GDP growth to the original level after increasing confidence and flexibility.

**Table 8.** Simulation of different policy scenarios for raising investment expectations due to the COVID-19 pandemic (unit: 10 million; %).

| | | Target I | | | Target II | | | Target III | | |
|---|---|---|---|---|---|---|---|---|---|---|
| | Initial Value | Policy I | Policy II | Policy III | Policy I | Policy II | Policy III | Policy I | Policy II | Policy III |
| GDP | 8,951,660 | 3.71% | 1.57% | 0.51% | 5.44% | 5.58% | 5.69% | 5.04% | 5.00% | 5.01% |
| Fixed Investment | 3,864,501 | 9.46% | 4.87% | 2.92% | 5.37% | 4.82% | 3.92% | 4.91% | 4.83% | 4.84% |
| Stock change | 393,442 | 7.36% | 4.22% | 2.99% | 5.49% | 3.45% | 3.99% | 4.99% | 4.94% | 4.485% |
| Investment Account: | | | | | | | | | | |
| Total | 4,257,943 | 14.67% | 4.88% | 2.93% | 5.38% | 5.39% | 3.93% | 4.91% | 4.82% | 4.81% |
| Household | 4,665,648 | 16.16% | 5.38% | 3.23% | 5.93% | 4.32% | 4.32% | 5.41% | 5.37% | 5.39% |
| Government | −277,332 | 39.61% | 13.25% | 7.93% | 14.51% | 36.85% | 10.51% | 13.18% | 12.82% | 12.46% |

Note: data were obtained from the authors' calculations.

In Target II, when confidence remains unchanged, a 3% tax cut and a 5.5% increase in spending are needed to achieve the 5.5% GDP growth target; when confidence declines, a more aggressive fiscal policy of a 5% tax cut and a 6% increase in spending can achieve the target; when confidence improves to its original level, a 3% tax cut and a 4% increase in spending are needed to achieve the target.

In Target III, when investment expectation remains unchanged, a fiscal policy of a 6.5% tax cut and a 5% increase in spending can achieve the target; when confidence falls, a 6.5% tax cut and a 6% increase in spending are needed to reach the target of 5% economic growth; when confidence rises and investment elasticity is improved, a 4% tax cut and a 5% increase in spending should be implemented at this time.

It is worth noting that the change in government investment in the investment account is higher than household investment under different fiscal policies. In particular, under Policy I in Target I, government investment grows higher than 10% when tax is reduced by 5% and spending is enlarged by 10%, with it reaching 39.61%. It can be seen that an active fiscal policy may lead to higher-than-average government investment. In addition, in the face of the downward pressure of the economy due to the COVID-19 pandemic, although fiscal policy can effectively mitigate the economic growth shock, it poses a greater challenge to government investment, and even if the investment is excessive. It will have a "crowding out" effect on household investment and it affects the social distribution structure.

4.2.3. Raising Import and Export Expectations

In terms of import and export trade, due to the impact of the COVID-19 pandemic, countries' production and consumption shrank, intercountry trade activities decreased significantly, and the economic recession was accompanied by a significant decrease in import and export expectations. The target policy mix used in raising import and export expectations is the same as the policy scenario for consumption, and the effects on GDP, imports, and exports under different policy scenarios are shown in Table 9.

**Table 9.** Simulation of different policy scenarios for raising import and export expectations due to the COVID-19 pandemic (unit: 10 million; %).

| | | Target I | | | Target II | | | Target III | | |
|---|---|---|---|---|---|---|---|---|---|---|
| | Initial Value | Policy I | Policy II | Policy III | Policy I | Policy II | Policy III | Policy I | Policy II | Policy III |
| GDP | 8,951,660 | 5.94% | 2.69% | 0.73% | 5.44% | 5.45% | 5.76% | 5.04% | 5.02% | 5.00% |
| Import | −1,649,280 | 8.87% | 4.00% | 2.89% | 5.41% | 5.91% | 4.01% | 4.94% | 3.82% | 4.96% |
| Export | 1,628,915 | 8.72% | 4.93% | 2.96% | 5.45% | 5.98% | 3.97% | 5.03% | 4.76% | 4.97% |
| Foreign Account | 1,663,569 | 8.92% | 3.19% | 2.99% | 5.49% | 5.98% | 4.01% | 5.03% | 4.96% | 5.14% |

Note: data were obtained from the authors' calculations.

In order to restore economic growth to its original level, after comparing the three policies in Target I, Policy III is most consistent with the Target I setting, when GDP growth grew by 0.73%, and imports and exports grew by 2.99% and 2.96%, respectively. In order to reach the target of 5.5% economic growth in Target II, the import and export confidence is set to remain unchanged, fall, and return to the original level under the three expected scenarios. At this point, the three confidence expectations are 5.49%, 5.98%, and 4.01% for

imports, and 5.45%, 5.91%, and 3.97% for exports. To meet the expectation of 5% GDP growth in Target III, the three confidence expectations are 5.03%, 3.82%, and 4.96% for imports and 4.94%, 4.76%, and 4.97% for exports. It can be seen that the above fiscal policies increase the GDP growth rate and stimulate imports and exports. Thus, the policy effects are obvious.

At the same time, it should be noted that the above types of fiscal policies make the overall level of exports higher than the level of imports. In 2021, China's net export growth rate was 773.67%, which is 5.95 times than the net export growth rate of 130.01% in 2020. Due the impact of the COVID-19 pandemic, on the one hand, raw materials for enterprises were lost and exports stagnated, but on the other hand, the blockage of commodity ports made it difficult to ship commodities out of the country, and stocks of commodities piled up, all of which hindered exports to some extent. Therefore, when implementing fiscal policy, it is important to focus on the points of economic growth and effectively drive the growth of imports and exports.

*4.3. Summary*

Through the above simulations of different policy scenarios for raising consumption, investment, import and export expectations, we found the following results.

First, the proactive fiscal policy has a significant pulling effect on raising consumption, investment, import, and export expectations. Under the three different scenario targets, the GDP growth rate rebounded to the original levels, 5.5% and 5%. One sees that fiscal policy can effectively mitigate the economic shock caused by the COVID-19 pandemic.

Second, in terms of residential consumption, the above fiscal policy can stimulate residential consumption to a certain extent, and the change in rural consumption is higher than urban consumption. The rural market has great potential, so we should further stimulate rural consumption, increase the investment in agricultural infrastructure, and at the same time, produce a good combination and optimize the income distribution structure to enhance the rural consumption capacity. Combined with the previous subsection, the weakening of urban consumption directly affects consumer services such as transportation, storage and postal, accommodation and dining, while the weakening of rural consumption directly affects primary industries such as food and tobacco, and agriculture. Urban residents have great potential for consumption demand for services, and the key to tap into is to improve the level of supply, namely effective supply.

Third, in terms of investment, the impact of the COVID-19 pandemic on investment is weaker than that of residential consumption, and the above fiscal policy will strongly reverse investment expectations and improve domestic demand. Combined with the previous subsection, it can be ascertained that, influenced by the economic scale contraction, for the main investment body of the real estate industry investment motivation is low, corporate financing willingness is weak, the real estate market continues to run at a low level. Therefore, the implementation process of fiscal policy should correctly play its positive guiding role to effectively drive the market investment demand back up.

Fourth, in terms of imports and exports, the above fiscal policy can effectively boost the demand for imports and exports, and the impact of the fiscal policy on exports is greater than that of imports. At present, China's export market share still maintains its advantage. As of January–March 2022, China's exports compared to the United States, Japan and South Korea grew significantly and had further momentum, so the implementation of favorable fiscal policy would further improve the share of China's exports. It is also noted that imports and exports expectations are significantly more affected by the COVID-19 pandemic than investment and less affected by consumption, and the changes are all greater than the changes in GDP.

## 5. Conclusions and Policy Suggestions

### 5.1. Conclusions

The following conclusions are drawn from this study.

First, Keynesian closure is valid, and the impact of the COVID-19 pandemic on China's economy is significant, with varying degrees of change in each expected indicator. The impact of the COVID-19 pandemic on GDP, urban consumption, rural consumption, investment, imports, and exports is at a rate of 2.35%, 7.96%, 9.79%, 4.10%, −3.13%, and 6.15%, respectively, and the impact of the COVID-19 pandemic is particularly severe on rural consumption. In addition, the marginal propensity to consume in urban and rural areas demonstrates a reverse and positive pull on consumption expectation, respectively; the investment elasticity demonstrates a positive pull on investment expectations, and the Armington elasticity of substitution and CET elasticity play a positive role in import and export expectations, respectively.

Second, the impact of the COVID-19 pandemic on different industries expectations is apparent on all sides, but there is a non-equilibrium. From the viewpoint of consumption demand, urban and rural residents have the largest changes in consumption demand for the tertiary and primary industries, respectively, and the overall change in consumption for rural residents is greater than that of urban residents. From the perspective of investment demand, the unbalanced impact of the COVID-19 pandemic on the three industries is reflected in the most significant changes in the tertiary industry, followed by the primary industry. In terms of import demand, the impact of the expected weakening on industry imports is concentrated in the secondary industry. In terms of industry types, the sectors of electrical machinery and equipment, oil and gas extraction products, and woodworking and furniture show more damage. From the point of view of export demand, the expected weakening of exports is manifested mainly in the primary industry, followed by the secondary industry. Food, energy, and timber imports are more deeply affected by the expected weakening. External uncertainty affects China's import and export trade, which plays a hindering role in positive economic growth.

Third, fiscal policies such as reducing taxes, lowering fees, and raising government fiscal spending are effective in mitigating the economic shock of the COVID-19 pandemic and help to achieve the new economic growth target. Among the three economic growth targets set, three expected confidence scenarios are set. In addition, when expectations are lower and the GDP growth target is higher, a more active fiscal policy is needed. In addition, fiscal policy has different effects on imports and exports while significantly boosting import and export trade. Fiscal policy has a greater impact on exports than imports, with exports being more deeply affected by direct supply-side shocks, while imports are mainly influenced by the demand side of the population. The contraction of resident consumption causes a significant decline in imports, and the demand-side changes have a greater pull on imports than the government's fiscal policy.

### 5.2. Policy Suggestions

Based on the above conclusions, we propose the following policy suggestions.

First, in terms of market expectations, counter-cyclical adjustment and cross-cyclical synergy should hedge the weakening trend of market expectations and cope with the impact of the COVID-19 pandemic in a calm manner. The weakening of market expectations is a phased development trend of the economic growth cycle. The organic combination of cross-cycle and counter-cyclical macroeconomic adjustment policies can effectively hedge the weakening trend of market expectations, help the economic system to give full play to the rebound effect, and cope with the impact of the COVID-19 pandemic in a calm manner. Above all, multiple measures boost consumer spending. It is necessary to guide deposit interest rates down to ease residents' savings and promote residents' steady consumption. At the same time, we encourage consumer credit services to support residents' consumption. Then, we will distribute cash to the low-income group and government consumption vouchers to the middle-income group. Furthermore, the strategic

base point is to stabilize investment and optimize domestic demand management policies. Studies show that the COVID-19 pandemic has a more pronounced expected impact on domestic demand. Financial and social capital should be used to promote the synergistic growth of traditional investment and investment in new industries. Last but not least, we will pay close attention to the situation of the COVID-19 pandemic and the impact of the response measures on imports and exports. Meanwhile, we should promote trade liberalization, expand the scale of credit to foreign trade enterprises, support enterprises to innovate foreign trade systems and markets, and maintain our position in the global industrial chain.

Second, in terms of industry impact, we will increase our efforts to support industries that are more affected by the COVID-19 pandemic and implement a package of relief and assistance policies for small- and medium-sized enterprises and individual entrepreneurs. To relieve the financial pressure of enterprises, governments at all levels have taken multiple measures to set up special funds to relieve the difficulties of small- and medium-sized enterprises, focusing on supporting industries that have been hit hard by the COVID-19 pandemic, especially consumer services such as accommodation, catering and tourism. On the one hand, the government guides take-out companies and other Internet platform companies to further reduce the standard of merchant service fees in the catering industry to reduce the operating costs of relevant catering enterprises. On the other hand, the government continues to implement the temporary travel agency deposit refund support policy, whereby the eligible travel agencies can apply for a temporary refund of the ratio increased by 100% and make up the deposit period and appropriate limit extension.

Third, in terms of macro policy, fiscal policy is needed to increase regulation and precise policy to effectively make up for the lack of social demand and promote sustained and steady economic recovery. Fiscal policy is the main policy direction of force, through a stronger fiscal policy to support the growth of infrastructure investment to create more domestic demand power. Overall, the policy intensity of fiscal revenue and expenditure needs to be further increased as the COVID-19 pandemic has impacted both ends of public finance revenue and expenditure, and the gap between revenue and expenditure has further widened. In addition, to reduce taxes and fees and increase transfer payments, fiscal policy should focus on key areas and segments by increasing the issuance of special treasury bonds and special bonds, increasing the level of deficit and other incremental tools to be used to fill the fiscal gap caused by tax cuts, cost increases, and pressure relief.

**Author Contributions:** Conceptualization, J.F.; methodology, X.Z.; software, J.F.; validation, H.W., and X.Z.; formal analysis, H.W.; investigation, J.F.; resources, J.F.; data curation, J.F.; writing—original draft preparation, J.F.; writing—review and editing, H.W.; visualization, H.W.; supervision, X.Z.; project administration, J.F.; funding acquisition, J.F. All authors have read and agreed to the published version of the manuscript.

**Funding:** This research was funded by Major Program of National Social Science Foundation of China grant number 21&ZD081.

**Institutional Review Board Statement:** Not applicable.

**Informed Consent Statement:** Not applicable.

**Data Availability Statement:** Data will be available if necessary.

**Conflicts of Interest:** The authors declare no conflict of interest.

## Appendix A

**Table A1.** Forty-two sectoral elasticity parameters table.

| Sector | LESELAS | | PRODELAS | $\rho_c^q$ | $\rho_c^t$ |
|--------|---------|-------|----------|-----------|-----------|
|        | Urban | Rural | | | |
| Sector 1 | 1.100 | 0.900 | 0.750 | 3.000 | 0.937 |
| Sector 2 | 0.900 | 1.100 | 0.500 | 0.750 | 0.937 |
| Sector 3 | 0.900 | 1.100 | 0.500 | 0.750 | 0.937 |
| Sector 4 | 0.900 | 1.100 | 0.500 | 0.750 | 0.937 |
| Sector 5 | 1.300 | 1.100 | 0.500 | 0.750 | 0.937 |
| Sector 6 | 1.037 | 1.500 | 0.900 | 1.500 | 1.500 |
| Sector 7 | 1.037 | 1.271 | 0.900 | 1.500 | 1.500 |
| Sector 8 | 1.037 | 1.271 | 0.900 | 1.500 | 1.500 |
| Sector 9 | 1.037 | 1.271 | 0.900 | 1.500 | 1.500 |
| Sector 10 | 1.037 | 1.271 | 0.900 | 1.500 | 1.500 |
| Sector 11 | 1.037 | 1.271 | 0.900 | 1.500 | 1.500 |
| Sector 12 | 1.037 | 1.271 | 0.900 | 1.500 | 1.500 |
| Sector 13 | 1.037 | 1.271 | 0.900 | 1.500 | 1.500 |
| Sector 14 | 1.037 | 1.271 | 0.900 | 1.500 | 1.500 |
| Sector 15 | 1.037 | 1.271 | 0.900 | 1.500 | 1.500 |
| Sector 16 | 1.037 | 1.271 | 0.900 | 1.500 | 1.500 |
| Sector 17 | 1.037 | 1.271 | 0.900 | 1.500 | 1.500 |
| Sector 18 | 1.037 | 1.271 | 0.900 | 1.500 | 1.500 |
| Sector 19 | 1.037 | 1.271 | 0.900 | 1.500 | 1.500 |
| Sector 20 | 1.037 | 1.271 | 0.900 | 1.500 | 1.500 |
| Sector 21 | 1.037 | 1.271 | 0.900 | 1.500 | 1.500 |
| Sector 22 | 1.037 | 1.271 | 0.900 | 1.500 | 1.500 |
| Sector 23 | 1.037 | 1.271 | 0.900 | 1.500 | 1.500 |
| Sector 24 | 1.037 | 1.271 | 0.900 | 1.500 | 1.500 |
| Sector 25 | 1.037 | 1.271 | 0.900 | 1.500 | 1.500 |
| Sector 26 | 1.037 | 1.271 | 0.900 | 1.500 | 1.500 |
| Sector 27 | 1.037 | 1.271 | 0.900 | 1.500 | 1.500 |
| Sector 28 | 1.037 | 1.271 | 0.900 | 1.500 | 1.500 |
| Sector 29 | 1.037 | 1.271 | 0.900 | 1.500 | 1.500 |
| Sector 30 | 0.923 | 0.677 | 1.249 | 0.500 | 0.375 |
| Sector 31 | 0.923 | 0.677 | 1.249 | 0.500 | 0.375 |
| Sector 32 | 0.923 | 0.677 | 1.249 | 0.500 | 0.375 |
| Sector 33 | 0.923 | 0.677 | 1.249 | 0.500 | 0.375 |
| Sector 34 | 0.923 | 0.677 | 1.249 | 0.500 | 0.375 |
| Sector 35 | 0.923 | 0.677 | 1.249 | 0.500 | 0.375 |
| Sector 36 | 0.923 | 0.677 | 1.249 | 0.500 | 0.375 |
| Sector 37 | 0.923 | 0.677 | 1.249 | 0.500 | 0.375 |
| Sector 38 | 0.923 | 0.677 | 1.249 | 0.500 | 0.375 |
| Sector 39 | 0.923 | 0.677 | 1.249 | 0.500 | 0.375 |
| Sector 40 | 0.923 | 0.677 | 1.249 | 0.500 | 0.375 |
| Sector 41 | 0.923 | 0.677 | 1.249 | 0.500 | 0.375 |
| Sector 42 | 0.923 | 0.677 | 1.249 | 0.500 | 0.375 |

Note: the 42 sectors in Appendix A correspond to the sectors in Table 5, respectively.

## Appendix B

**Table A2.** Forty-two benchmark sectoral elasticity parameters calibration table.

| Sector | LESELAS | | PRODELAS | $\rho_c^q$ | $\rho_c^t$ |
|--------|---------|-------|----------|-----------|-----------|
|        | Urban | Rural | | | |
| Sector 1 | 1.375 | 0.675 | 0.250 | 1.500 | 0.469 |
| Sector 2 | 1.125 | 0.825 | 0.167 | 0.375 | 0.469 |
| Sector 3 | 1.125 | 0.825 | 0.167 | 0.375 | 0.469 |
| Sector 4 | 1.125 | 0.825 | 0.167 | 0.375 | 0.469 |

**Table A2.** *Cont.*

| Sector | LESELAS | | *PRODELAS* | $\rho_c^q$ | $\rho_c^t$ |
|---|---|---|---|---|---|
| | **Urban** | **Rural** | | | |
| Sector 5 | 1.625 | 0.825 | 0.167 | 0.375 | 0.469 |
| Sector 6 | 1.296 | 1.125 | 0.300 | 0.750 | 0.750 |
| Sector 7 | 1.296 | 0.953 | 0.300 | 0.750 | 0.750 |
| Sector 8 | 1.296 | 0.953 | 0.300 | 0.750 | 0.750 |
| Sector 9 | 1.296 | 0.953 | 0.300 | 0.750 | 0.750 |
| Sector 10 | 1.296 | 0.953 | 0.300 | 0.750 | 0.750 |
| Sector 11 | 1.296 | 0.953 | 0.300 | 0.750 | 0.750 |
| Sector 12 | 1.296 | 0.953 | 0.300 | 0.750 | 0.750 |
| Sector 13 | 1.296 | 0.953 | 0.300 | 0.750 | 0.750 |
| Sector 14 | 1.296 | 0.953 | 0.300 | 0.750 | 0.750 |
| Sector 15 | 1.296 | 0.953 | 0.300 | 0.750 | 0.750 |
| Sector 16 | 1.296 | 0.953 | 0.300 | 0.750 | 0.750 |
| Sector 17 | 1.296 | 0.953 | 0.300 | 0.750 | 0.750 |
| Sector 18 | 1.296 | 0.953 | 0.300 | 0.750 | 0.750 |
| Sector 19 | 1.296 | 0.953 | 0.300 | 0.750 | 0.750 |
| Sector 20 | 1.296 | 0.953 | 0.300 | 0.750 | 0.750 |
| Sector 21 | 1.296 | 0.953 | 0.300 | 0.750 | 0.750 |
| Sector 22 | 1.296 | 0.953 | 0.300 | 0.750 | 0.750 |
| Sector 23 | 1.296 | 0.953 | 0.300 | 0.750 | 0.750 |
| Sector 24 | 1.296 | 0.953 | 0.300 | 0.750 | 0.750 |
| Sector 25 | 1.296 | 0.953 | 0.300 | 0.750 | 0.750 |
| Sector 26 | 1.296 | 0.953 | 0.300 | 0.750 | 0.750 |
| Sector 27 | 1.296 | 0.953 | 0.300 | 0.750 | 0.750 |
| Sector 28 | 1.296 | 0.953 | 0.300 | 0.750 | 0.750 |
| Sector 29 | 1.296 | 0.953 | 0.300 | 0.750 | 0.750 |
| Sector 30 | 1.154 | 0.508 | 0.167 | 0.250 | 0.188 |
| Sector 31 | 1.154 | 0.508 | 0.167 | 0.250 | 0.188 |
| Sector 32 | 1.154 | 0.508 | 0.167 | 0.250 | 0.188 |
| Sector 33 | 1.154 | 0.508 | 0.167 | 0.250 | 0.188 |
| Sector 34 | 1.154 | 0.508 | 0.167 | 0.250 | 0.188 |
| Sector 35 | 1.154 | 0.508 | 0.167 | 0.250 | 0.188 |
| Sector 36 | 1.154 | 0.508 | 0.167 | 0.250 | 0.188 |
| Sector 37 | 1.154 | 0.508 | 0.167 | 0.250 | 0.188 |
| Sector 38 | 1.154 | 0.508 | 0.167 | 0.250 | 0.188 |
| Sector 39 | 1.154 | 0.508 | 0.167 | 0.250 | 0.188 |
| Sector 40 | 1.154 | 0.508 | 0.167 | 0.250 | 0.188 |
| Sector 41 | 1.154 | 0.508 | 0.167 | 0.250 | 0.188 |
| Sector 42 | 1.154 | 0.508 | 0.167 | 0.250 | 0.188 |

Note: the 42 sectors in Appendix B correspond to the sectors in Table 5, respectively.

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
