# Peer review of "A General Equilibrium Analysis of Achieving the Goal of Stable Growth by China’s Market Expectations in the Context of the COVID-19 Pandemic"

_sustainability, doi:10.3390/su142215072_

Round 1

Reviewer 1 Report

This paper applies a CGE model to simulate the impact of COVID-19 on market agents as well as industry sectors. I think the topic is of practical meanings for economic development in China in the long run, particularly in the current age full of uncertainty, risk, and crisis. I provide the following revision advices to improve the paper:

1.        The literature review is recommended to be modified and strengthened. More international papers published in high-quality journals are recommended to be included, which helps a better generalization of previous studies.

2.        The contribution of this study should be better stated. “To find a macroeconomic theory” is quite an ambitious objective to be achieved by this paper.

3.        Please unify the names of the epidemic (COVID-19, new crown epidemic, the new crown pneumonia epidemic, etc.) throughout the paper.

4.        The English of this paper needs to be improved. A native speaker is highly recommended to help to improve the language.

Author Response

Dear Reviewer,

Thanks very much for taking your time to review this manuscript. I really appreciate all your comments and suggestions! Please find my itemized responses in below and my paper in the re-submitted files.

Comments to the author:

  1. a) The literature review is recommended to be modified and strengthened. More international papers published in high-quality journals are recommended to be included, which helps a better generalization of previous studies.

Response to this question: We are grateful to this reviewer’s suggestion. We have revised and enhanced the literature review by citing some high-quality international journal papers and adding some international literature reviews on the impact of the COVID-19 pandemic on other aspects of the macroeconomy to make the literature review in this paper more comprehensive and complete.

  1. b) The contribution of this study should be better stated. “To find a macroeconomic theory” is quite an ambitious objective to be achieved by this paper.

Response to this question: Thank this reviewer for this suggestion. Through macro closure analysis, comparing exogenous shocks with real data, seeking macroeconomic theories that are more in line with the interpretation of reality is an important part of this study. In terms of the choice of macroscopic closure in 3.1.2, we focus on Keynesianism, the macroeconomic theory chosen in this study, from two aspects of the reasons and characteristics of choosing Keynesianism, and explain in the form of footnotes that Keynesianism is more from the perspective of demand, and China's fiscal and monetary policies focus on the views represented by Keynesianism.

  1. c) Please unify the names of the epidemic (COVID-19, new crown epidemic, the new crown pneumonia epidemic, etc.) throughout the paper.

Response to this question: Thank this reviewer for this suggestion, this paper has unified the epidemic into COVID-19 pandemic.

  1. d) The English of this paper needs to be improved. A native speaker is highly recommended to help to improve the language.

Response to this question: Thank this reviewer for this suggestion. We have carefully checked grammatical mistakes and misspelling and corrected them one by one. In addition, English language and style have been modified accordingly.

Thanks again for your suggestions. We have addressed the comments and the amendments are highlighted in red in the revised manuscript. We hope that the revision is acceptable, and I look forward to hearing from you soon. 

Reviewer 2 Report

I found the topic of the paper interesting. However, I have the following concerns/comments:

a) I missed a brief literature review. Although the authors provide insights about their motivation and the background behind their work, I missed a discussion about other relevant works that deal with the respective field. This should be done either as a separate section or in one or two paragraphs in the introduction.

b) Figures 1 and 2 are unclear. It would be useful if the authors let the GDP growth rate to "hit" on the right hand side axis, since it seems that is rather stable, which is not true, as consumption and investment growth rates have a larger variability.

c) In page 2, line 65, the authors claim that  "some political figures in developed countries led by the United States continue to wreak havoc on the global value chain and suply chain". I am not sure what the authors mean at this point and what is the added value of this statement. I believe that the authors should explain in more detail this point.

d) In page 2, the authors categorize the weaking of expectations. They distinguish as separate factors investment expectations (second) and import and export and foreign investment expectations (third). Why FDI expectations do not included in investment expectations?

e) In section 3.1.2, the authors explain why they choose the Keynesian closure, providing three key characteristics that are related to the outbreak of the pandemic. Although I agree with the characteristics mentioned in this section, I think that the authors should explain how the Keynesian closure captures the supply-side disruptions emerged since the outbreak of the pandemic. Is the Keynesian closure siutable to capture supply-side effects or it focuses only on demand aspects?

f) In section 3.2.1 there is no combination of a simultaneous shrinkage on both Urban and Rural consumption expectations. I believe that the authors should elaborate on this or at least explain why they don noy experimented with this change.

g) The language is weak and there are several typos. For example, in the abstract in lines 20-21 I missed the verb in the sentence. In line 25, whta is the difference between reducing taxes and lowering taxes? In page 3, in lines 105-108 the authors repeat twice the assumptions of the neoclassical school, etc.

Author Response

Dear Reviewer,

Thanks very much for taking your time to review this manuscript. I really appreciate all your comments and suggestions! Please find my itemized responses in below and my paper in the re-submitted files.

Comments to the author:

  1. a) I missed a brief literature review. Although the authors provide insights about their motivation and the background behind their work, I missed a discussion about other relevant works that deal with the respective field. This should be done either as a separate section or in one or two paragraphs in the introduction.

Response to this question: We are grateful to this reviewer’s suggestion. Regarding this suggestion to add a discussion involving other relevant works in the respective field, we have reviewed previous studies and added three literature reviews in the introduction, which discuss the impact of the COVID-19 pandemic on the labor, financial markets, and business demand, respectively, making the literature review more comprehensive in terms of the impact of the COVID-19 pandemic on the respective field.

  1. b) Figures 1 and 2 are unclear. It would be useful if the authors let the GDP growth rate to "hit" on the right hand side axis, since it seems that is rather stable, which is not true, as consumption and investment growth rates have a larger variability.

Response to this question: Thank this reviewer for this suggestion. In the previous study, we put GDP, consumption and investment growth rates in the same line chart, but ignored the problem that GDP and consumption and investment are not of the same order of magnitude, which would cause the change of GDP to appear stable intuitively, even closed to the X axis, compressing the salience of GDP growth rate. Therefore, in the modification of Figures 1 and 2, we show the GDP growth rate in the form of histogram, consumption and investment growth rates in the form of line chart, so that we can more intuitively feel the change range of various indicators.

  1. c) In page 2, line 65, the authors claim that "some political figures in developed countries led by the United States continue to wreak havoc on the global value chain and suply chain". I am not sure what the authors mean at this point and what is the added value of this statement. I believe that the authors should explain in more detail this point.

Response to this question: We refer to the statement of the Ministry of Foreign Affairs of China in this paragraph, which means that some countries acted independently on the basis of national security zones, which had a great impact on global value chains and supply chains. At the same time, referring to Figure 1. World Uncertainty Index (WUI) over time in《The World Uncertainty Index》, it is not difficult to see that the world risk index brought by the Sino US trade war is higher than other events, and the developed countries led by the United States have damaged the global value chain. In fact, the formation and development of the global industrial chain and supply chain should be the result of the joint action of market rules and enterprise choices. We have revised this sentence and thank this reviewer for this suggestion.

  1. d) In page 2, the authors categorize the weaking of expectations. They distinguish as separate factors investment expectations (second) and import and export and foreign investment expectations (third). Why FDI expectations do not included in investment expectations?

Response to this question: Thank this reviewer for this suggestion. In fact, the third point only includes the import and export expectations, not the foreign investment expectation, which is a part of the investment expectation. At present, the global epidemic continues to spread, and the instability and uncertainty of the supply chain and industrial chain are also further rising. In order to cope with the following foreign investment work, China also proposes to further increase the efforts to attract foreign capital and attract more investment from multinational companies.

  1. e) In section 3.1.2, the authors explain why they choose the Keynesian closure, providing three key characteristics that are related to the outbreak of the pandemic. Although I agree with the characteristics mentioned in this section, I think that the authors should explain how the Keynesian closure captures the supply-side disruptions emerged since the outbreak of the pandemic. Is the Keynesian closure siutable to capture supply-side effects or it focuses only on demand aspects?

Response to this question: The reason for choosing Keynesianism is that for China, both fiscal policy and monetary policy focus on the Keynesian perspective. The core of Keynesianism revolves around two assumptions: one is that the currency is non neutral, and the other is that the market is imperfect. Keynesianism is more from the demand side, while the supply chain is from the supply side. This paper does not mention the content of the supply chain, and we will simulate and authenticate the supply side’s impact on resource mismatch in future research.

  1. f) In section 3.2.1 there is no combination of a simultaneous shrinkage on both Urban and Rural consumption expectations. I believe that the authors should elaborate on this or at least explain why they don noy experimented with this change.

Response to this question: Thank this reviewer for this suggestion. In previous studies, our consumption sensitivity analysis was mainly to analyze whether urban consumption or rural consumption changed more, so we chose one kind of change and the other kind of unchanged analysis method, without considering the scenario of both shrinking at the same time. According to your suggestion, we have added the situation of simultaneous reduction of marginal propensity to consume in urban and rural to 3.2.1, that is, Change 1:URBANHH reduced by 1.25 times, RURALHH reduced by 1.25 times.

  1. g) The language is weak and there are several typos. For example, in the abstract in lines 20-21 I missed the verb in the sentence. In line 25, whta is the difference between reducing taxes and lowering taxes? In page 3, in lines 105-108 the authors repeat twice the assumptions of the neoclassical school, etc.

Response to this question: Thank this reviewer for this suggestion. We have carefully checked grammatical mistakes and misspelling and corrected them one by one. In addition, English language and style have been modified accordingly.

Thanks again for your suggestions. We have addressed the comments and the amendments are highlighted in red in the revised manuscript. We hope that the revision is acceptable, and I look forward to hearing from you soon. 

Reviewer 3 Report

First of all, thank the author(s) for your hard work. However, I think the following comments will make it better and more robust.

-The title can be made shorter and more concise.

- The use of abbreviations is not consistent throughout the paper. For example, GDP in the abstract. Please use the abbreviation once being introduced.

- The researcher(s) should pay attention to the research gap that is still not sufficient. Therefore, please add more arguments related to the research gap in the introduction. 

-Please add one or two paragraphs on the impact of the COVID-19 pandemic on the industry sectors various in general and the on-the-market agents especially. 

-Also, please avoid long paragraphs (e.g.,  paragraphs 2 and 4  in the introduction).

- As far as I have seen,  part "3. Calibration of macroscopic closure and benchmark parameters" is too long. Therefore, please try to restructure by only giving short information.

-Please try to restructure part 5. Policy Suggestions in a separate section on research implications. The current writing is not well structured.

-Along the same lines, it is necessary to mention the research conclusion, limitations and recommendations in a separate section. 

-Finally, I think the article will benefit from proofreading since grammatical mistakes and misspelling is common in the article. For example,  631 The research........, The researchers. 

I hope that my comments can help you to improve your manuscript

Author Response

Dear Reviewer,

Thanks very much for taking your time to review this manuscript. I really appreciate all your comments and suggestions! Please find my itemized responses in below and my paper in the re-submitted files.

Comments to the author:

  1. a) The title can be made shorter and more concise.

Response to this question: The article title has been changed from A General Equilibrium Analysis of Effective Paths to Achieve China’s Stable Growth Goals in the Context of the Century Epidemic - A Perspective on Stabilizing Market Expectations into A General Equilibrium Analysis of Achieving the Goal of Stable Growth by China’s Market Expectations in the Context of the COVID-19 pandemic.

  1. b) The use of abbreviations is not consistent throughout the paper. For example, GDP in the abstract. Please use the abbreviation once being introduced.

Response to this question: Thank this reviewer for this suggestion, the article has been revised to define some abbreviations for the first time, such as GDP, CGE, SAM, etc.

  1. c) The researcher(s) should pay attention to the research gap that is still not sufficient. Therefore, please add more arguments related to the research gap in the introduction. 

Response to this question: We are grateful to this reviewer’s suggestion. We note the still insufficient research gap and include three paragraphs in the introduction on the impact of the COVID-19 pandemic on labor, financial markets, and business demand aspects to complete this paper's literature review on the macroeconomic impact of the COVID-19 pandemic.

  1. d) Please add one or two paragraphs on the impact of the COVID-19 pandemic on the industry sectors various in general and the on-the-market agents especially. 

Response to this question: Thank this reviewer for this suggestion. With regard to the proposal to add one or two paragraphs on the impact of the COVID-19 pandemic on the industry sectors various in general, we have added a new section of the overall impact of the COVID-19 pandemic on the industry in 3.3. As for the impact of market agents, the market agents understood in this paper is the main body of consumption, investment, import and export. We have discussed the impact of consumption, investment, import and export on various sectors in 3.3.

  1. e) Also, please avoid long paragraphs (e.g.,  paragraphs 2 and 4  in the introduction).

Response to this question: The article has been revised to avoid too long comments in reviewer paragraphs. For example, in the second paragraph of the introduction, the four types of expectations are divided into paragraphs, and in the original fourth paragraph of the introduction, the views of different macroeconomic schools are formed into a separate paragraph.

  1. f) As far as I have seen,  part "3. Calibration of macroscopic closure and benchmark parameters" is too long. Therefore, please try to restructure by only giving short information.

Response to this question: On the choice of macroscopic closure, the reasons and characteristics of the choice are described respectively. On this basis, some irrelevant discussions are deleted to make the information shorter. In the calibration of benchmark parameters, the benchmark parameters of consumption, import and export are too long, so various changes are tabulated to make the information more concise.

  1. g) Please try to restructure part 5. Policy Suggestions in a separate section on research implications. The current writing is not well structured.

Response to this question: In view of the content and structure of Part 5, the title of Part 5 is revised as Conclusions and Policy Suggestions, which are divided into a separate section.

  1. h) Along the same lines, it is necessary to mention the research conclusion, limitations and recommendations in a separate section. 

Response to this question: Similarly, the research conclusions have been discussed in a separate section.

  1. Finally, I think the article will benefit from proofreading since grammatical mistakes and misspelling is common in the article. For example,  631 The research........, The researchers. 

Response to this question: Thank this reviewer for this suggestion. We have carefully checked grammatical mistakes and misspelling and corrected them one by one. In addition, English language and style have been modified accordingly.

Thanks again for your suggestions. We have addressed the comments and the amendments are highlighted in red in the revised manuscript. We hope that the revision is acceptable, and I look forward to hearing from you soon. 

Round 2

Reviewer 2 Report

The authors tried to address all my comments and suggestions and their revised manuscript is significanlty improved compared to the previous one. Thus, my recommendation is to accept the paper in present form.

Reviewer 3 Report

Thank you for revising the manuscript thoroughly and addressing many of my concerns. The manuscript has been improved a lot. Good job.